# SCFormer: Spatial Coordination for Efficient and Robust Vision Transformers

## Abstract

We investigate the design of visual backbones with a focus on optimizing both efficiency and robustness. While recent advancements in hybrid Vision Transformers (ViTs) have significantly enhanced efficiency, achieving state-of-the-art performance with fewer parameters, their robustness against domain-shifted and corrupted inputs remains a critical challenge. This trade-off is particularly difficult to balance in lightweight models, where robustness often relies on wider channels to capture diverse spatial features. In this paper, we present SCFormer, a novel hybrid ViT architecture designed to address these limitations. SCFormer introduces Spatial Coordination Attention (SCA), a mechanism that coordinates cross-spatial pixel interactions by deconstructing and reassembling spatial conditions with diverse connectivity patterns. This approach broadens the representation boundary, allowing SCFormer to efficiently capture more diverse spatial dependencies even with fewer channels, thereby improving robustness without sacrificing efficiency. Additionally, we incorporate an Inceptional Local Representation (ILR) block to flexibly enrich local token representations before self-attention, enhancing both locality and feature diversity. Through extensive experiments, SCFormer demonstrates superior performance across multiple benchmarks. On ImageNet-1K, SCFormer-XS achieves 2.5% higher top-1 accuracy and 10% faster GPU inference speed compared to FastViT-T8. On ImageNet-A, SCFormer-L (30.1M) surpasses RVT-B (91.8M) in robustness accuracy by 5.6% while using $3\times$ fewer parameters. These results underscore the effectiveness of our design in achieving a new state-of-the-art balance between efficiency and robustness.

## 1 Introduction

Recent progress in computer vision has led to a paradigm shift from convolutional neural networks (ConvNets) (Liu et al., 2022; He et al., 2016; Szegedy et al., 2016) to Vision Transformers (ViTs) (Dosovitskiy et al., 2020) and their hybrid variants (Wu et al., 2022; Liu et al., 2021; Li et al., 2022; Pan et al., 2022). Unlike ConvNets, which primarily focus on local pixel processing using fixed-sized filters, ViTs utilize self-attention (SA) mechanisms that enable dynamic interactions across both short- and long-range spatial dependencies. This has allowed ViTs to excel in capturing complex, non-local relationships in images, granting them superior performance on a wide range of computer vision tasks. However, the high-dimensionality of image data, coupled with the ViT's reliance on global pixel relationships, poses significant computational and efficiency challenges, especially during the initial self-attention calculations. This has made ViTs computationally intensive and parameter-heavy, limiting their broader deployment in real-world applications such as edge computing or autonomous systems, where both efficiency and robustness are crucial.

To address these challenges, research has shifted towards optimizing SA for more efficient visual learning. Recent strategies involve reducing SA's internal dimensions via pooling or convolution-based downsampling (Wu et al., 2022; Li et al., 2022), integrating convolution layers within SA computations (Vasu et al., 2023a; Wang et al., 2021), or refining SA to enhance local token interactions (Shaker et al., 2023; Pan et al., 2022). These methods have led to the development of hybrid ViTs that fuse the inductive biases of ConvNets with the flexibility of SA, achieving impressive results in terms of accuracy and parameter efficiency. However, despite these advances, a significant gap remains in the robustness of lightweight ViT and other efficient vision models, particularly in challenging test scenarios involving domain shifts, adversarial attacks, or noisy data.

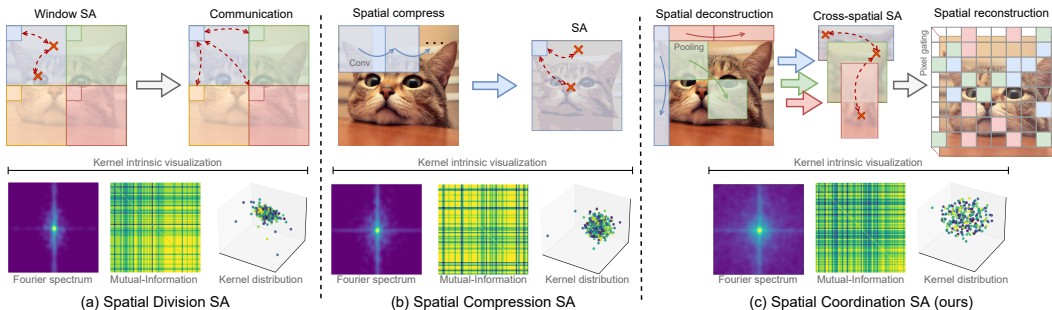

Figure 1: Depiction and corresponding kernel visualization of two existing efficient SA schemes: spatial division (a) (Liu et al., 2021), spatial compression (b) (Wang et al., 2022), and the proposed spatial coordination SA (c). In (a) and (b), pixel dependencies are restricted to individual spatial maps with fixed connectivity. In contrast, our approach explores cross-spatial pixel coordination with dynamic connectivity. The visualization shows that the proposed SA scheme captures richer frequency-level information (Fourier spectrum) with reduced channel-level information redundancy (mutual information), leading to a broader representation boundary in the embedding space (kernel distribution). We offer detailed implementations and more visualizations in appendix A.1-A.3.

For instance, while FastViT-SA24 (Vasu et al., 2023a) achieves higher accuracy than ConvNeXt-S (Liu et al., 2022) on the ImageNet-1K dataset with fewer parameters (20M vs. 49M), its performance on robustness benchmarks such as ImageNet-R and ImageNet-SK lags behind. This robustness-efficiency tradeoff presents a critical challenge, especially for lightweight architectures that need to generalize well across diverse tasks without expanding their parameter budgets. As modern applications such as autonomous driving, medical imaging, and mobile vision increasingly rely on high-efficiency models, addressing this gap is more important than ever.

We identify two primary paradoxes in existing ViT designs that contribute to this robustness gap: (1) robustness in lightweight architectures is closely tied to channel width, with wider channels offering greater capacity to capture diverse spatial features, such as textures and frequency patterns, which are crucial for handling domain shifts (Liu et al., 2023; Mao et al., 2022); and (2) existing efficient self-attention (SA) designs, which leverage locality priors, often restrict the spatial representation capacity of the model. Specifically, spatially divided (Liu et al., 2021) and spatially reduced SA schemes (Yu et al., 2022; Shaker et al., 2023) (Fig. 1 (a) and (b)) enforce fixed local pixel connections that can lead to a loss of cross-channel information and prevent the model from fully utilizing the global context. These limitations hinder the ability of lightweight ViTs to generalize well across diverse and corrupted input data.

In this work, we propose a new architecture, the Spatial Coordination Transformer (SCFormer), which aims to address these robustness challenges by rethinking how locality priors and spatial diversity are incorporated into ViTs. Our key innovations are twofold: First, we decouple locality enrichment from the attention block by introducing an Inceptional Local Representation (ILR) block. Unlike traditional convolution layers, which impose fixed spatial dependencies, the ILR block flexibly captures a wide range of local frequency information via inception-like convolution operations before each attention block. This flexible locality induction enriches the token representations with diverse high-level features, preparing them for more effective attention processing. The inception mechanism allows the model to dynamically adjust to varying spatial scales, leading to improved robustness and feature diversity across different spatial patterns. Second, we introduce Spatial Coordination Attention (SCA), a novel approach that breaks the conventional depth-wise processing of SA. Rather than focusing exclusively on individual spatial maps, SCA dynamically coordinates pixel interactions across different spatial maps with varying connectivity patterns. By leveraging multiple pooling descriptors, we deconstruct spatial conditions and reassemble them as substrates for global coordination through SA. This process enables the model to maintain a more diverse set of spatial interactions, enhancing its ability to generalize across tasks with limited channel budgets. After spatial coordination, a pixel gating operation reconstructs the original spatial maps, efficiently propagating cross-spatial coordination scores while preserving the semantic integrity of the output.

In Fig. 1, we show how SCA outperforms mainstream efficient SAs (division (Liu et al., 2021) and compression (Wang et al., 2022)) on principle metrics, such as the spatial Fourier spectrum and channel mutual information. SCA achieves a richer representation of frequency information

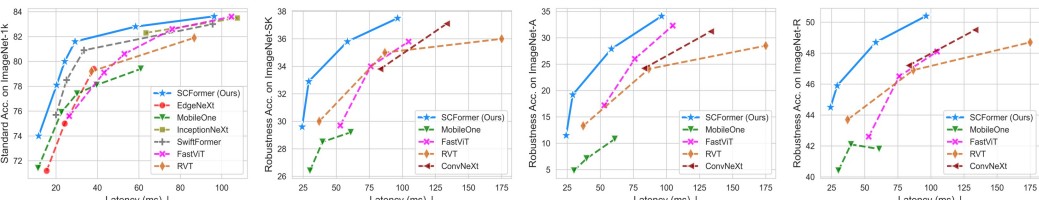

Figure 2: Performance comparison on ImageNet-1K, -SK, -A, and -R. The proposed SCFormer achieves superior trade-off among standard accuracy, latency, and robustness over existing models.

and greater channel feature diversity, resulting in a broader representation boundary. These key properties, in practice, enable SCFormer to greatly outperform existing leading backbones in both robustness and clean accuracy (Fig. 2). In summary, our contributions are: **(1)** We introduce *SCFormer*, a novel hybrid ViT architecture designed for robust and efficient visual learning. **(2)** We propose Spatial Coordination Attention (SCA), which facilitates cross-spatial pixel coordination to broaden the representation boundary and improve robustness with fewer channels. **(3)** We introduce the Inceptional Local Representation (ILR) block, which flexibly enriches local token representations before self-attention, enhancing both locality and feature diversity. **(4)** Extensive experiments on image classification, dense prediction, and cross-domain tasks demonstrate that SCFormer consistently sets new benchmarks, achieving a superior trade-off between efficiency and robustness.

## 2 RELATED WORK

**Efficient CNNs.** They are tailored for practical applications. Operators such as Depthwise Separable Convolution (DWConv) (Chollet, 2017) and Group Convolution (Ioannou et al., 2017) have been pivotal in developing streamlined architectures, leading to the creation of lightweight and rapid models like MobileNets (Howard et al., 2017; Sandler et al., 2018), ShuffleNets (Zhang et al., 2018; Ma et al., 2018), GhostNet (Han et al., 2020), and TVConv (Chen et al., 2022). These models, by capitalizing on filter redundancy within visual patterns, have carved out a niche of efficient models extensively applied in edge computing scenarios. Subsequent endeavors have harnessed architecture search to build a network like EfficientNets (Tan & Le, 2019; 2021). Simultaneously, research on pruning and compression has aimed to streamline large CNNs, optimizing both the number of parameters and computational load. Recently, the focus has shifted to efficient ViTs, noted for surpassing CNNs through superior long-range pixel dependency learning. Nonetheless, some design principles of efficient CNNs, including DWConv, remain integral to cutting-edge ViTs. In this work, we harness multi-view insights from InceptionNets (Szegedy et al., 2016) to enhance the local representation (Conv) blocks within our hybrid ViT architecture.

**Efficient ViTs.** Most existing efficient ViTs (Vasu et al., 2023a; Li et al., 2022; Pan et al., 2022; Shaker et al., 2023) employ hybrid architectures. Previous works (Liu et al., 2021; Wang et al., 2021) introduce Convs to perform patch merging and spatial downsampling, reforming the isotropic architecture of ViT in a pyramidal style. To further pursue efficiency, recent work focuses on combining Conv operators (Li et al., 2022; Shaker et al., 2023; Pan et al., 2022) within SA mechanisms to reduce complexity and running latency. The key is to use local operators, such as DWConv, to foster information overlap between individual tokens/patches prior to SA computation. This approach can reduce the inner dimension of SA to reduce the complexity and also introduce visual inductive biases (e.g., locality) into SA for efficient visual modeling. The MetaFormer (Yu et al., 2022) summarized modern SA designs as the token mixer and used a pooling alternative to validate its viewpoint. Unlike existing token-mixing attentions, this paper presents a novel SCA that peeks at local spatial features from multiple cross-channel views to promote local information coordinates across different filters. It enriches spatial condition representation to efficiently promote discriminative and robust visual representation learning.

**Robustness Designs.** Some prior research investigated the robustness of vision backbones (Mao et al., 2022; Liu et al., 2023; Hendrycks et al., 2021b; Wang et al., 2019). RVT (Mao et al., 2022) studies the relationship between robustness and architecture designs in ViT frameworks. By simply combining robustness designs, it proposes a robust vision transformer that achieves favorable performance on various robust benchmarks. Afterward, ConvNeXt (Liu et al., 2022) has also implicitly improved the robustness of the vision backbone by using fewer operators and a shallower depth to trade for greater channel widths. There is also a comprehensive study (Liu et al., 2023) on robust-

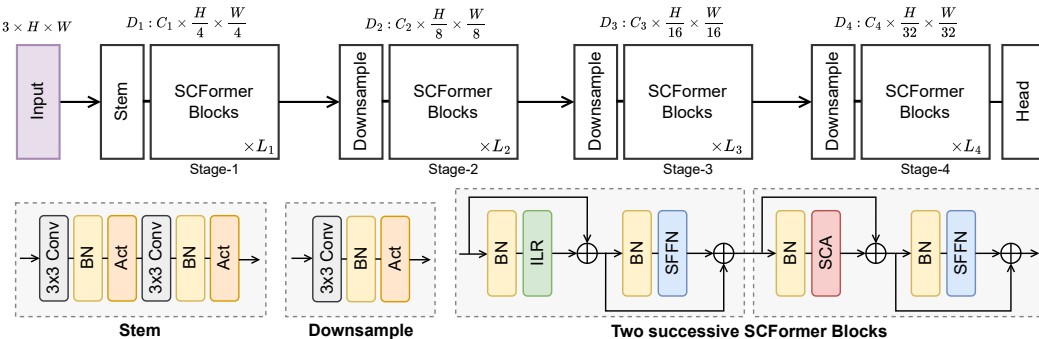

Figure 3: Overview of SCFormer architecture. Each set of two successive SCFormer blocks is configured with the ILR in the first block and the SCA in the second block.

ness that reveals a trade-off between natural robustness and general precision. Most of the existing backbones (Mao et al., 2022; Liu et al., 2022; Vasu et al., 2023a) are essentially making trade-offs in operator and architecture designs, making it hard to obtain robustness of parameter efficiency and consistency of performance in both robust and general tasks. In this paper, we propose the SCA that explores the coordination of local spatial conditions to see more robustness-related patterns from fewer channels, thereby overriding the tradeoff issue.

## 3 METHODOLOGY

The overall architecture of SCFormer is presented in Fig. 3. In the following, we first introduce the SCA in §3.1. Then, ILR, SFFN, and the overall configurations of SCFormer are discussed in §3.2.

### 3.1 SPATIAL COORDINATION ATTENTION

This subsection introduces the SCA (Fig. 4 (d)) as a fundamental technique for spatial modeling. In modern ViTs (Shaker et al., 2023; Wang et al., 2021; Liu et al., 2021), the input feature $z_{in} \in \mathbb{R}^{c \times h \times w}$ undergoes token mixing, using operators such as DWconv or SA for depth-wise modeling. These focus on intra-spatial information exchange but limit channel-wise interactions, which helps to emphasize key spatial patterns. However, this also restricts cross-channel expression, which is vital for maintaining recognition robustness when test data deviates from training. To overcome this, current token mixers require significantly wider channels, increasing parameter demands.

To improve the robustness-efficiency trade-off, we propose a novel SCA. It enhances spatial diversity by coordinating pairwise relationships among diverse local conditions across channels, enriching the conditions' representation. Then, SCA applies a pixel gating mechanism for token mixing with spatial reconstruction, utilizing coordinated features for robust visual representation. Specifically, we first deconstruct spatial conditions, as shown in Fig. 4 (c), where the input 2D features are summarized by three pooling descriptors ($d$) covering distinct local regions.

$$z_w = \text{Flatten}(d_w(z_{in}))\text{W}_w, \ z_h = \text{Flatten}(d_h(z_{in}))\text{W}_h, \ z_s = \text{Flatten}(d_s(z_{in}))\text{W}_s,$$
$$z_{rc} = \text{LayerNorm}(\text{Concat}(\{z_w, z_h, z_s\})). \tag{1}$$

In Eq. (1), we first repeatedly describe the regional spatial condition of $z_{in}$ in local width ($d_w$), height ($d_h$), and square ($d_s$) regions. Three linear projections ($\text{W}_w, \text{W}_h, \text{W}_s$) are used subsequently to enhance their representation, respectively. Afterward, we concatenate these outputs along channels as the regional condition map $z_{rc} \in \mathbb{R}^{n_d \times c_{sub}}$. It reveals the diversified spatial conditions of $z_{in}$, which are implicitly encoded in the different pixel relationships present in the original feature. The $z_{rc}$ also benefits from a reduced length compared to $z_{in}$, allowing faster SA computation.

The process in Eq. (1) deconstructs different regional spatial conditions from the input feature. As shown in Fig. 4 (a)-left, by processing the same feature with different pooling descriptors $d_w$, $d_h$, and $d_s$, spatial conditions vary in 2D representations. We further visualize discrete distributions of these described results $z_w$, $z_h$, and $z_s$ in 3D space by t-SNE in Fig. 4 (a)-right. They have non-overlapped boundaries indicative of representing varied spatial conditions. This observation inspired

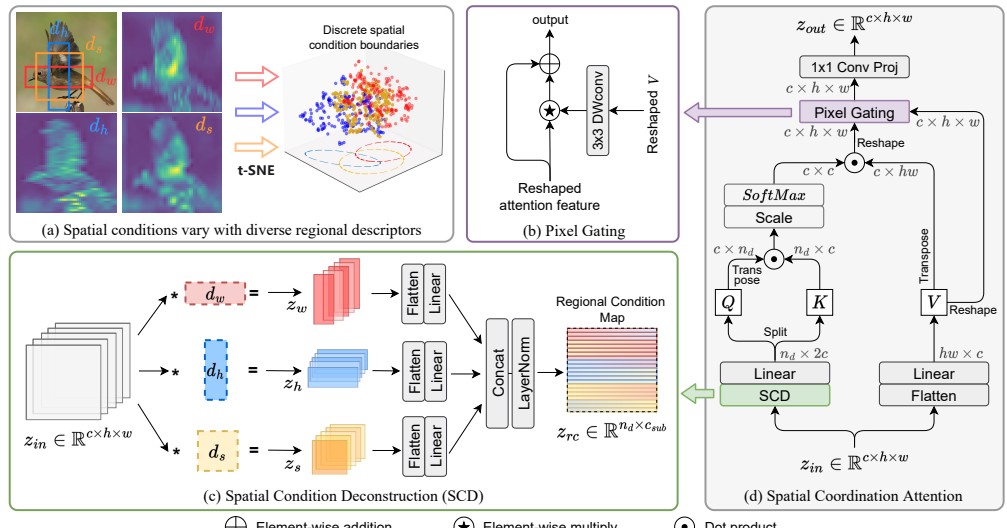

Figure 4: (a) We show that spatial conditions vary in 2D representation and have distinct boundaries in the embedding space (Van der Maaten & Hinton, 2008) when processed by different pooling descriptors. (b) The pixel gating for mixing tokens with structure-prior knowledge. (c) The procedure of spatial condition deconstruction before attention calculation. (d) The architecture of SCA.

us to treat $z_w$, $z_h$, and $z_s$ as a group of substrates (integrated into $z_{rc}$) and leverage SA to formulate pairwise synthetic relations among each of their embedding features, thereby efficiently generating richer spatial conditions beyond the original representation.

Based on the regional condition map $z_{rc}$ and input $z_{in}$, we then introduce the spatial coordination attention. To enable the coordination between regional conditions captured from different spatial maps (channels), we compute the $Q$, $K$ from $z_{rc}$ and $V$ from $z_{in}$, respectively.

$$Q, K = \text{Split}(z_{rc}\text{W}_{qk}), \ \ V = z_{in}\text{W}_v, \tag{2}$$

here we use a linear layer ($\text{W}_{qk}$) to expand the embedding (channel) dimension of $z_{rc}$ from $c_{sub}$ to $2c$ and Split it half-and-half as $Q$ and $K$. Then, unlike existing token mixers that perform attention on tokens, we compute the attention map along the embedding dimension similar to (Ali et al., 2021). This is for building pairwise correspondence between local regional conditions across channels. We then act the attention on transposed V for activating different combinations of spatial conditions. In the following, we omit the concept of multi-head (Dosovitskiy et al., 2020) for simplicity.

$$x_{att} = SoftMax(\frac{Q^T \cdot K}{t}) \cdot V^T, \tag{3}$$

where $t$ is a learnable temperature to scale the inner products before softmax. In Eq. (3), we compute the attention along the channel to build a pairwise correspondence between the spatial conditions and reweight each token. After the above information exchange and coordination across channels, we then use the pixel gating operation with residual branch to reconstruct the spatial information:

$$x_{spg} = x_{att} + x_{att} * \text{DWconv}(V), x_{out} = \text{PWconv}(x_{spg}). \tag{4}$$

In Eq. (4) we omit the dimension reshape for simplicity; "$*$" indicates the element-wise multiply operation for pixel-to-pixel gating; $x_{out}$ is the output of SCA. Instead of directly applying a local Conv on $x_{att}$ for mixing tokens, we extract the pixel positional information from the structure preserved V using a $3\times3$ DWconv and then perform pixel-wise gating on $x_{att}$ for spatial reconstruction with structure priors. Afterward, we use a residual add branch to preserve the prior information and a PWconv for the final projection. The feature variations in SCA is visualized in appendix A.3.

## 3.2 HYBRID ARCHITECTURES

Here, we first present the Inception Local Representation (ILR) block, designed to introduce locality priors before the SCA module. In addition, we discuss the selective incorporation of DWconv within the feed-forward network of our architecture to enhance efficiency. Lastly, we provide an overview of the general architecture and configurations of our SCFormers.

**ILR.** Drawing inspiration from the inceptionNet (Szegedy et al., 2016), we use the inception thinking to introduce the locality priors while boosting the spatial diversity before SCA calculation.

As shown in Fig.5 (a), for an input feature $z_{in} \in \mathbb{R}^{c \times h \times w}$, we first divide it into three segments along the channel dimension, each segment with the channel number of $c_\phi$, $c_\alpha$ and $c_\beta$, respectively. They are then processed by distinct DWconvs with kernel sizes of 7×7, 3×1, and 1×3, respectively. The outcomes are concatenated to form the output. This approach enables nuanced local representation modeling, introducing the locality while enriching the spatial information's variety and scale.

**SFFN.** Incorporating a DWconv within FFN is a popular strategy. Prior studies (Wang et al., 2022; Wu et al., 2022) place the DWconv between the two projection layers (1×1 Conv) to process spatial information at higher dimensions, which improves scene parsing but brings complexity overload. Recent works (Vasu et al., 2023a; Shaker et al., 2023) place the DWconv before the first projection to speed up the operation. However, this can lead to suboptimal performance due to the preliminary encoding of spatial details at lower dimensions.

This paper proposes to selectively incorporate the in-FFN DWConv for better efficiency. We observe that the efficacy of the in-FFN DWconv in encoding spatial information is significant in the initial network stages, where spatial information is plentiful. However, its computational

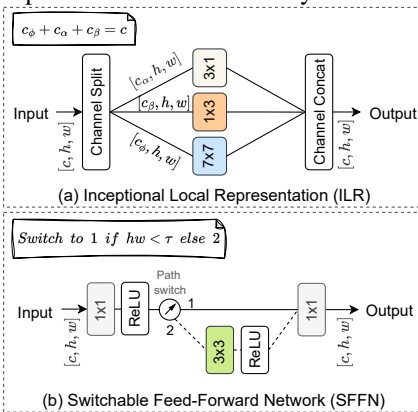

(a) Inceptional Local Representation (ILR)

(b) Switchable Feed-Forward Network (SFFN)

Figure 5: Block designs for ILR on (a), and SFFN on (b).

demand spikes in the later stages due to the greatly increased channel width. Therefore, deploying the in-FFN DWconv uniformly is inefficient, as its utility diminishes in later phases, instead contributing to computational burden. Thus, we use the switchable FFN (SFFN) at Fig. 5 (b), which only activates the in-FFN DWconv in the early network stages (spatial size $(h*w) >$ the control parameter $\tau$). This allows for adaptable adjustment of using the in-FFN DWconv for better efficiency.

**Configurations.** The SCFormer configurations are shown in Tab. 1. Detailed architecture and hyperparameters are discussed in the appendix. We present fixed configurations for all variants. Specifically, we set the number of attention heads as {1,2,5,8} for the four stages. Given the input feature $z_{in} \in \mathbb{R}^{c \times h \times w}$ for current SCformer block, the divided channels $c_\phi$, $c_\alpha$, and $c_\beta$ in all the InceptLP blocks are set to be $\frac{1}{2}c$, $\frac{1}{4}c$, and $\frac{1}{4}c$, respectively; To cope with the spatial dimensions in different stages, the kernel sizes of local pooling descriptors $\{d_w,$

Table 1: Configuration of six SCFormer variants. #Channels: number of channels per stage; #Blocks: number of SCFormer blocks per stage; #$\tau$: the switch parameter in SFFN; $HW$ means the product of the input image height and width.

| Variants | #Channels | #Blocks | FLOPs | #Params |
|---|---|---|---|---|
| -XXS | [24,48,120,192] | [2,2,4,2] | 0.3G | 2.0M |
| -XS | [32,64,160,256] | [2,2,6,2] | 0.6G | 3.9M |
| -S | [40,80,200,320] | [2,2,8,2] | 1.0G | 6.7M |
| -M | [48,96,200,384] | [2,2,10,4] | 1.4G | 11.8M |
| -ML | [64,128,300,512] | [2,4,12,4] | 3.4G | 22.9M |
| -L | [72,144,320,512] | [4,4,16,4] | 5.2G | 31.4M |

$d_h, d_s\}$ in SCA are set to be $\{[12,6], [6,12], [8,8]\}$, $\{[8,4], [4,8], [4,4]\}$, $\{[6,3], [3,6], [2,2]\}$, and $\{[3,1], [1,3], [1,1]\}$ for spatial condition deconstruction in the stage-1, 2, 3, and 4, respectively.

## 4 EXPERIMENTS

We evaluate SCFormer on standard / robust image classification tasks (§4.1), object detection and segmentation tasks (§4.2), cross-domain retrieval tasks (§4.3). Finally, we conduct ablation studies to show the robustness roadmap (§4.4) and give activation visualization on OOD samples (§4.5).

### 4.1 CLASSIFICATION ON IMAGENET-1K AND ROBUST BENCHMARKS

**Setup for ImageNet-1k.** The ImageNet-1k dataset (Deng et al., 2009) consists of 1.3M training and 50K validation samples. To ensure a fair comparison, we train our SCFormer following standard ViT training protocols (Touvron et al., 2021). Specifically, the models are trained for 300 epochs using the AdamW optimizer, with a peak learning rate of 2e-3 and a total batch size of 2048. The warmup period lasts for 5 epochs, and the learning rate is decayed using a cosine schedule. All

Table 2: Comparison on ImageNet-1k classification. All the latency and throughput are measured using one 2080ti GPU, which may differ from some official results for hardware variations. ∗ and † marks denote models using architecture search and reparameterization, respectively.

| Model | Eval image size | Param (M) | FLOPs (G) | Latency (ms) ↓ | Throughput (fps) ↑ | Top-1 Acc (%) ↑ |
|---|---|---|---|---|---|---|
| EdgeNeXt-XXS (Maaz et al., 2022) | 256 | 1.3 | 0.3 | 15.5 | 2070 | 71.2 |
| MobileOne-S0† (Vasu et al., 2023b) | 224 | 2.1 | 0.3 | 11.3 | 2979 | 71.4 |
| SkipAT-T (Venka. et al., 2024) | 224 | 5.8 | 1.1 | 14.9 | 2213 | 72.9 |
| SCFormer-XXS | 224 | 2.0 | 0.3 | 11.6 | 2874 | **74.0** |
| EdgeNeXt-XS (Maaz et al., 2022) | 256 | 2.3 | 0.5 | 24.2 | 1322 | 75.0 |
| FastViT-T8† (Vasu et al., 2023a) | 256 | 3.6 | 0.7 | 26.4 | 1210 | 75.6 |
| SwiftFormer-XS (Shaker et al., 2023) | 224 | 3.5 | 0.6 | 20.0 | 1604 | 75.7 |
| MobileOne-S1† (Vasu et al., 2023b) | 224 | 4.8 | 0.8 | 22.6 | 1415 | 75.9 |
| SCFormer-XS | 224 | 3.9 | 0.7 | 20.3 | 1458 | **78.1** |
| MobileOne-S2† (Vasu et al., 2023b) | 224 | 7.8 | 1.3 | 30.1 | 1006 | 77.4 |
| SwiftFormer-S (Shaker et al., 2023) | 224 | 6.1 | 1.0 | 25.1 | 1316 | 78.5 |
| EfficientNet-B1∗ (Tan & Le, 2019) | 256 | 7.8 | 0.7 | 45.6 | 702 | 79.1 |
| FastViT-T12† (Vasu et al., 2023a) | 256 | 6.8 | 1.4 | 43.2 | 744 | 79.1 |
| EdgeNeXt-S (Maaz et al., 2022) | 256 | 5.6 | 1.3 | 38.2 | 843 | 79.4 |
| SCFormer-S | 224 | 6.7 | 1.0 | 24.2 | 1334 | **80.0** |
| PoolFormer-S12 (Yu et al., 2022) | 224 | 11.9 | 1.8 | 31.0 | 1008 | 77.2 |
| MobileOne-S3† (Vasu et al., 2023b) | 224 | 10.1 | 1.9 | 39.6 | 808 | 78.1 |
| RVT-Ti (Mao et al., 2022) | 224 | 10.9 | 1.3 | 37.1 | 860 | 79.2 |
| MobileOne-S4† (Vasu et al., 2023b) | 224 | 14.8 | 3.0 | 60.9 | 525 | 79.4 |
| FastViT-SA12† (Vasu et al., 2023a) | 256 | 10.9 | 1.9 | 53.0 | 604 | 80.6 |
| SwiftFormer-L1 (Shaker et al., 2023) | 224 | 12.1 | 1.6 | 33.5 | 955 | 80.9 |
| SCFormer-M | 224 | 11.8 | 1.5 | 29.2 | 1175 | **81.6** |
| SkipAT-S (Venka. et al., 2024) | 224 | 22.1 | 4.0 | 88.4 | 351 | 80.2 |
| Swin-T (Liu et al., 2021) | 224 | 29.0 | 4.5 | 90.0 | 352 | 81.3 |
| PoolFormer-S36 (Yu et al., 2022) | 224 | 31.0 | 5.0 | 86.9 | 368 | 81.4 |
| RVT-S (Mao et al., 2022) | 224 | 23.3 | 4.7 | 86.7 | 370 | 81.9 |
| ConvNeXt-T (Liu et al., 2022) | 224 | 29.0 | 4.0 | 83.1 | 413 | 82.1 |
| FasterViT-0 (Hatamizadeh et al., 2024) | 224 | 31.4 | 3.3 | 69.2 | 519 | 82.1 |
| InceptionNeXt-T (Yu et al., 2023) | 224 | 29.0 | 4.2 | 63.4 | 546 | 82.3 |
| FastViT-SA24† (Vasu et al., 2023a) | 256 | 20.6 | 3.8 | 76.1 | 446 | 82.6 |
| SCFormer-ML | 224 | 22.9 | 3.5 | 58.4 | 603 | **82.8** |
| SkipAT-B (Venka. et al., 2024) | 224 | 86.7 | 15.2 | 241.7 | 128 | 82.2 |
| PoolFormer-M48 (Yu et al., 2022) | 224 | 73.0 | 11.6 | 182.8 | 175 | 82.5 |
| RVT-B (Mao et al., 2022) | 224 | 91.8 | 17.7 | 175.3 | 180 | 82.7 |
| SwiftFormer-L3 (Shaker et al., 2023) | 224 | 28.5 | 4.0 | 94.6 | 360 | 83.0 |
| ConvNeXt-S (Liu et al., 2022) | 224 | 50.0 | 8.7 | 133.4 | 281 | 83.1 |
| FasterViT-1 (Hatamizadeh et al., 2024) | 224 | 53.4 | 5.3 | 98.4 | 340 | 83.2 |
| Swin-B (Liu et al., 2021) | 224 | 88.0 | 15.4 | 233.9 | 136 | 83.5 |
| InceptionNeXt-S (Yu et al., 2023) | 224 | 49.0 | 8.4 | 107.5 | 293 | 83.5 |
| EfficientNet-B5∗ (Tan & Le, 2019) | 456 | 30.0 | 9.9 | 463.0 | 69 | **83.6** |
| FastViT-SA36† (Vasu et al., 2023a) | 256 | 30.4 | 5.6 | 99.1 | 326 | **83.6** |
| SCFormer-L | 224 | 31.4 | 5.2 | 96.3 | 355 | **83.6** |

training and testing images are resized to $224 \times 224$. Training is conducted using PyTorch on 8 NVIDIA A100 GPUs. Detailed settings and distillation results are provided in the Appendix.

**Setup for Robust Benchmarks.** The robustness is assessed on ImageNet-C (IN-C) (Hendrycks & Dietterich, 2019), -R (Hendrycks et al., 2021a), -SK (Wang et al., 2019), and -A (Hendrycks et al., 2021b). These datasets are commonly used to evaluate classification robustness against out-of-distribution, corrupted, and adversarial samples. Following (Liu et al., 2022; Mao et al., 2022), we report our performance by directly testing the ImageNet-1k trained model on these datasets.

**Comparison on ImageNet-1k.** In Tab. 2, we compare SCFormer with the latest SOTA models on ImageNet-1k. Without using architecture search (AS) or reparameterization (REP), SCFormer achieves a superior accuracy-speed tradeoff. Compared to FastViT-T8 (Vasu et al., 2023a), which leverages REP, SCFormer-XS improves top-1 accuracy by 2.5% with 10% faster inference. Additionally, SCFormer-M surpasses SwiftFormer-L1 (Shaker et al., 2023) with 0.7% higher accuracy and 10% faster speed. Our larger variants, SCFormer-ML and SCFormer-L, significantly outperform recent SOTAs in accuracy-efficiency tradeoff, using simpler requirements and smaller input sizes. Notably, SCFormer-L achieves 83.6% top-1 accuracy with reduced latency and faster speed compared to AS-based EfficientNet-B5 (Tan & Le, 2019) and REP-based FastViT-SA36 (Vasu et al., 2023a), validating the efficacy of SCA and other components for efficient visual learning.

**Comparison on Robust Benchmarks.** In Tab. 3, we assess our model's robustness on several benchmarks. All SCFormer variants achieve top performance across these tests. Despite RVT (Mao et al., 2022) being tailored for robustness, SCFormer outperforms it. Notably, SCFormer-L (31.4M) surpasses RVT-B (91.8M) by 5.6% on IN-A and 1.5% on IN-SK, using only 1/3 of the parameters.

Table 3: Evaluation on robustness benchmarks. We report the mean corruption error (lower is better) for ImageNet-C (IN-C) and top-1 accuracy for other datasets. The latency metrics are the same as Tab. 2, which we omit there for simplicity.

| Model | Params | FLOPs | IN-1K (↑) | IN-C (↓) | IN-A (↑) | IN-R (↑) | IN-SK (↑) |
|---|---|---|---|---|---|---|---|
| MobileOne-S0 (Vasu et al., 2023b) | 2.1 | 0.3 | 71.4 | 86.4 | 2.3 | 32.9 | 19.3 |
| EdgeNeXt-XXS (Maaz et al., 2022) | 1.3 | 0.3 | 71.2 | 86.8 | 2.6 | 30.0 | 18.5 |
| SCFormer-XXS | 2.0 | 0.3 | **74.0** | **84.5** | **4.1** | **33.6** | **22.1** |
| MobileOne-S1 (Vasu et al., 2023b) | 4.8 | 0.8 | 75.9 | 80.4 | 2.7 | 36.7 | 22.6 |
| EdgeNeXt-XS (Maaz et al., 2022) | 2.3 | 0.5 | 75.0 | 81.7 | 4.6 | 33.0 | 22.0 |
| SCFormer-XS | 3.9 | 0.7 | **78.1** | **77.1** | **6.6** | **38.4** | **27.3** |
| MobileOne-S2 (Vasu et al., 2023b) | 7.8 | 1.3 | 77.4 | 73.6 | 4.8 | 40.4 | 26.4 |
| EdgeNeXt-S (Maaz et al., 2022) | 5.6 | 1.3 | 79.4 | 74.5 | 7.7 | 39.9 | 26.1 |
| SCFormer-S | 6.7 | 1.0 | **80.0** | **72.8** | **11.5** | **44.5** | **29.6** |
| MobileOne-S3 (Vasu et al., 2023b) | 10.1 | 1.9 | 78.1 | 71.6 | 7.1 | 42.1 | 28.5 |
| MobileOne-S4 (Vasu et al., 2023b) | 14.8 | 3.0 | 79.4 | 68.1 | 10.8 | 41.8 | 29.2 |
| RVT-Ti (Mao et al., 2022) | 8.6 | 1.3 | 78.4 | 58.2 | 13.3 | 43.7 | 30.0 |
| FastViT-SA12 (Vasu et al., 2023a) | 10.9 | 1.9 | 80.6 | 62.2 | 17.2 | 42.6 | 29.7 |
| SCFormer-M | 11.8 | 1.5 | **81.6** | **55.3** | **19.2** | **45.9** | **32.9** |
| ConvNeXt-T (Liu et al., 2022) | 29.0 | 4.0 | 82.1 | 53.2 | 24.2 | 47.2 | 33.8 |
| RVT-S (Mao et al., 2022) | 22.1 | 4.7 | 81.7 | 50.1 | 24.1 | 46.9 | 35.0 |
| FastViT-SA24 (Vasu et al., 2023a) | 20.6 | 3.8 | 82.6 | 55.3 | 26.0 | 46.5 | 34.0 |
| Swin-T | 29.0 | 4.5 | 81.3 | 62.0 | 21.6 | 41.3 | 29.1 |
| SCFormer-ML | 22.9 | 3.5 | **82.8** | **49.0** | **27.9** | **48.7** | **35.8** |
| ConvNeXt-S (Liu et al., 2022) | 73.0 | 11.6 | 82.5 | 51.2 | 31.2 | 49.5 | 37.1 |
| RVT-B (Mao et al., 2022) | 91.8 | 17.7 | 82.7 | 46.8 | 28.5 | 48.7 | 36.0 |
| FastViT-SA36 (Vasu et al., 2023a) | 30.4 | 5.6 | **83.6** | 51.8 | 32.3 | 48.1 | 35.8 |
| SCFormer-L | 31.4 | 5.2 | **83.6** | **46.7** | **34.1** | **50.4** | **37.5** |

Similar gains are seen in other variants. SCFormer consistently excels in both robust and general benchmarks, demonstrating the effectiveness of our SCA in learning robust visual representations.

## 4.2 OBJECT DETECTION AND SEGMENTATION

We evaluate SCFormer on multiple dense prediction/scene parsing tasks using ImageNet-1k trained models. For object detection and instance segmentation, we use the MS-COCO dataset (Lin et al., 2014) with the Mask-RCNN framework (He et al., 2017), adhering to standard protocols (Vasu et al., 2023a) for fair comparisons. For semantic segmentation, we assess our models in the ADE20k dataset (Zhou et al., 2017) with the semantic FPN decoder, following established settings (Vasu et al., 2023a; Shaker et al., 2023) to ensure fairness.

Table 4: Results using SCFormer as the backbone on dense prediction tasks. We follow mainstream practices to use the Mask-RCNN framework with a $1\times$ training schedule for object detection and instance segmentation on the MS-COCO dataset (Lin et al., 2014). The semantic segmentation is performed on the ADE20K dataset (Zhou et al., 2017) with the semantic FPN decoder. The backbone latency is measured using the input image image size of $512\times512$.

| Backbone | Latency | Detection and Instance Segmentation | | | | | | Semantic |
|---|---|---|---|---|---|---|---|---|
| | | $AP^b$ | $AP^b_{50}$ | $AP^b_{75}$ | $AP^m$ | $AP^m_{50}$ | $AP^m_{75}$ | mIoU(%) |
| ResNet-50 (He et al., 2016) | 159.4 | 38.0 | 58.6 | 41.4 | 34.4 | 55.1 | 36.7 | 36.7 |
| PoolFormer-S12 (Yu et al., 2022) | 101.7 | 37.3 | 59.0 | 40.1 | 34.6 | 55.8 | 36.9 | 37.2 |
| FastViT-SA12 (Vasu et al., 2023a) | 118.9 | 38.9 | 60.5 | 42.2 | 35.9 | 57.6 | 38.1 | 38.0 |
| SwiftFormer-L1 (Shaker et al., 2023) | 108.0 | **41.2** | 63.2 | 44.8 | 38.1 | 60.2 | 40.7 | 41.4 |
| SCFormer-M | **91.1** | **41.2** | **63.5** | **45.7** | **38.4** | **60.7** | **41.2** | **41.7** |
| ResNet-101 | 187.3 | 40.0 | 60.6 | 44.0 | 36.1 | 57.5 | 38.6 | 38.8 |
| PoolFormer-S24 (Yu et al., 2022) | 195.2 | 40.1 | 62.2 | 43.4 | 37.0 | 59.1 | 39.6 | 40.3 |
| FastViT-SA24 (Vasu et al., 2023a) | 207.0 | 42.0 | 63.5 | 45.8 | 38.0 | 60.5 | 40.5 | 41.0 |
| SwiftFormer-L3 (Shaker et al., 2023) | 231.7 | 42.7 | 64.4 | 46.7 | 39.1 | 61.7 | 41.8 | **43.9** |
| SCFormer-ML | **185.4** | **42.8** | **64.7** | **47.1** | **39.2** | **61.9** | **42.3** | **43.9** |
| PoolFormer-S36 (Yu et al., 2022) | 290.2 | 41.0 | 63.1 | 44.8 | 37.7 | 60.1 | 40.0 | 42.0 |
| FastViT-SA36 (Vasu et al., 2023a) | 302.4 | 43.8 | 65.1 | 47.9 | 39.4 | 62.0 | 42.3 | 42.9 |
| SCFormer-L | **288.7** | **44.3** | **65.2** | **48.2** | **40.1** | **62.3** | **43.0** | **44.3** |

As shown in Tab. 4, SCFormer achieves state-of-the-art results on dense prediction/scene parsing tasks. SCFormer-L outperforms REP-based FastViT-SA36 (Vasu et al., 2023a) by 0.5%, 0.7%, and 1.4% in $AP^b$, $AP^m$, and mIoU, respectively, while reducing GPU latency. Furthermore, SCFormer-M surpasses SwiftFormer-L1 (Shaker et al., 2023) in all metrics, with a 10% speed advantage. These results highlight the effectiveness of our SCA and hybrid components in achieving an accurate and efficient visual representation without relying on REP or NAS.

Table 5: Comparison on two cross-domain retrieval datasets. We report the rank (r) = 1 accuracy and mean average precision (mAP), both higher the better.

| Backbone | Retrieve Time | SYSU-MM01 (Wu et al., 2017) | | | | LLCM (Zhang & Wang, 2023) | | | |
|---|---|---|---|---|---|---|---|---|---|
| | | All Search | | Indoor Search | | VIS to IR | | IR to VIS | |
| | | r=1 | mAP | r=1 | mAP | r=1 | mAP | r=1 | mAP |
| ResNet-50 (He et al., 2016) (AGW) | **1.0×** | 47.9 | 47.8 | 55.1 | 63.7 | 57.1 | 59.4 | 47.2 | 55.1 |
| ConvNeXt-T (Liu et al., 2022) | 1.5× | 53.9 | 51.1 | 62.4 | 64.3 | 59.2 | 60.3 | 49.1 | 56.4 |
| FastViT-SA24 (Vasu et al., 2023a) | 0.7× | 54.2 | 52.8 | 64.7 | 64.9 | 61.4 | 61.6 | 52.5 | 57.9 |
| PoolFormer-S36 (Yu et al., 2022) | 1.1× | 52.8 | 50.1 | 61.1 | 62.7 | 59.0 | 60.1 | 48.3 | 55.8 |
| InceptionNeXt-T (Yu et al., 2023) | 0.8× | 56.3 | 55.1 | 68.7 | 68.1 | 61.8 | 63.1 | 52.9 | 58.3 |
| SCFormer-ML | 0.7× | **58.0** | **57.0** | **71.3** | **69.5** | **62.6** | **65.7** | **54.1** | **60.1** |
| ConvNeXt-S (Liu et al., 2022) | 3.7× | 65.5 | **62.1** | 77.7 | 72.9 | 65.2 | **66.7** | 58.3 | 65.1 |
| SwiftFormer-L3 (Shaker et al., 2023) | 1.7× | 64.7 | 61.0 | 75.8 | 70.9 | 64.1 | 66.1 | 57.1 | 63.9 |
| FastViT-SA36 (Vasu et al., 2023a) | 1.8× | 65.1 | 61.8 | 76.2 | 71.1 | 64.8 | 65.9 | 57.6 | 64.2 |
| PoolFormer-M48 (Yu et al., 2022) | 4.9× | 65.4 | 62.0 | 78.1 | **72.6** | 64.2 | 65.8 | 57.7 | 64.1 |
| SCFormer-L | 1.7× | **65.7** | 61.8 | **78.5** | 72.4 | **65.4** | 66.5 | **58.9** | **65.2** |

## 4.3 CROSS-DOMAIN IMAGE RETRIEVAL

We evaluate our model on cross-domain retrieval tasks to test its ability to learn robust feature distances under challenging domain shifts and fine-grained sample complexities. As image retrieval depends on ranking feature similarities, robust distance metrics in the embedding space are key for accuracy. This evaluation goes beyond classification robustness, testing the model's capacity to distinguish fine nuances among highly similar samples. Specifically, one visible-infrared image retrieve (SYSU-MM01 (Wu et al., 2017) and one visible-lowlight image retrieve (LLCM (Zhang & Wang, 2023) datasets are used. For SYSU-MM01, we follow the standard protocol from (Ye et al., 2021), and for LLCM, we use the official protocol (Zhang & Wang, 2023). All results are obtained by alternating the backbone in the AGW Re-ID framework (Ye et al., 2021), standardizing the input dimensions to 224×224 for all models, except FastViT, which uses 256×256. Speed comparisons are based on relative retrieval times, with ResNet-50 (the default AGW backbone) as the time reference.

Results in Tab. 5 show that SCFormer outperforms state-of-the-art models for cross-domain fine-grained image understanding. Specifically, SCFormer-ML achieves 1.7% and 2.6% higher rank-1 accuracy than InceptionNeXt-T (Yu et al., 2023) in all-search and indoor-search modes of SYSU-MM01, respectively, with similar gains on LLCM. Furthermore, SCFormer-L surpasses ConvNeXt-S across all evaluation protocols for both datasets, with over 2× faster speed. These results confirm SCFormer's superior ability to learn consistent feature distances in cross-domain scenarios.

## 4.4 ABLATION ROADMAP ON ROBUSTNESS

We conduct ablation experiments to validate the efficacy of our proposed components. Using ConvNeXt-T (Liu et al., 2022) as the baseline, we gradually transformed it into SCFormer-ML. Each modification is evaluated on ImageNet-1k and four robust benchmarks, demonstrating the variations in accuracy and robustness. Results are presented in the Tab. 6.

Table 6: Upgrading the ConvNeXt-T progressively to SCFormer-ML. The blue-marked rows denote enhancement facilitated by our proposed components.

| Row | Modifications | Param (M) | Latency (ms) | IN-1K (↑) | IN-C (↓) | IN-A (↑) | IN-R (↑) | IN-SK (↑) |
|---|---|---|---|---|---|---|---|---|
| 0 | ConvNeXt-T (Baseline) | 29.0 | 83.1 | 82.1 | 53.2 | 24.2 | 47.2 | 33.8 |
| 1 | Width: [96,192,384,768] → [64,128,300,512] | 14.9 | 53.9 | 79.0 | 74.3 | 12.1 | 44.3 | 30.6 |
| 2 | In-block Norm: LN→BN | 14.9 | 45.4 | 79.0 | 74.2 | 12.1 | 44.4 | 30.6 |
| 3 | Replace the last 7x7 dw in each stage to SCA | 16.5 | 46.3 | 79.9 | 68.2 | 16.5 | 45.8 | 31.6 |
| 4 | Depth→[2,4,8,4]; replace all dw with ILR-SCA | 19.7 | 46.9 | 82.1 | 52.7 | 24.5 | 48.0 | 33.9 |
| 5 | Improvements to ConvNeXt-T (row.0) | 32% | 44% | – | 0.5% | 0.3% | 0.8% | 0.1% |
| 6 | Block shortcut: ConvNet Style → ViT Style | 19.7 | 48.3 | 82.2 | 52.4 | 24.6 | 48.2 | 34.0 |
| 7 | FFN/MLP → SFFN | 19.8 | 53.5 | 82.4 | 50.9 | 25.3 | 48.5 | 34.7 |
| 8 | Align Stem and PE layers to SCFormer | 20.4 | 54.7 | 82.5 | 50.1 | 25.5 | 48.5 | 35.0 |
| 9 | Depth: [2,4,8,4] → [2,4,12,4] (SCFormer-ML) | 22.9 | 58.4 | 82.8 | 49.0 | 27.9 | 48.7 | 35.8 |
| 10 | Improvements to ConvNeXt-T (row.0) | 21% | 30% | 0.7% | 4.2% | 3.7% | 1.5% | 1.3% |

**Preparation:** We first reduced the channel widths to [64,128,300,512] that aligns with our SCFormer-ML's configuration, leading to a significant drop in robust accuracy across all benchmarks, as the robustness of ConvNeXt-T heavily depends on using larger channel widths to mechan-

ically obtain more spatial conditions (see Tab. 6, rows 0-1). In row 2, we switch the LayerNorm within blocks to BatchNorm for also the alignment with our model's settings.

**Spatial Modeling:** By replacing the last $7\times7$ DWconv with SCA in each stage (row 3), an increase is observed in robustness with minimal latency gain, highlighting SCA's ability to efficiently learn robust visual representations from reduced channels. Substituting all DWconvs with ILR and SCA, and adjusting the number of blocks to even for successively using them, we significantly boosted the robustness further (row 4). Compared to the ConvNeXt-T, we reduce parameters/Latency by 32%/44%, while enhancing all robustness metrics.

**Channel Mixing:** We first change the ConvNet-like single shortcut to the ViT-like double shortcut within basic blocks (row 6). Then, we replace the FFN with our SFFN, which enhances performance on all the datasets with limited latency increases (row 7). This validates the effectiveness of SFFN's switchable scheme that uses the middle DWconv only in early network stages.

**Final Alignment:** Final modifications (rows 8-9) transition ConvNeXt-T to SCFormer-ML with architecture design modifications. In rows 1-10, we reduce the parameters by over 20% and running latency by 30%, while also significantly improving the performance on ImageNet-1k and four robustness benchmarks, demonstrating the efficiency and effectiveness of each proposed component.

### 4.5 ACTIVATION MAP VISUALIZATION ON GENERATED OOD SAMPLES

We visualize Grad-CAM (Selvaraju et al., 2017) activation maps for one ImageNet-1K sample ("normal cat") and two out-of-distribution (OOD) samples (anime, painting) generated by DALLE (Ramesh et al., 2022) using different style prompts. Swin-Tiny and PvTv2-b2 are selected to compare with our SCFormer-L; they are all trained on ImageNet-1k. In addition, we present the feature cosine similarity matrix for a more intuitive comparison. The visualizations are listed in Fig. 6.

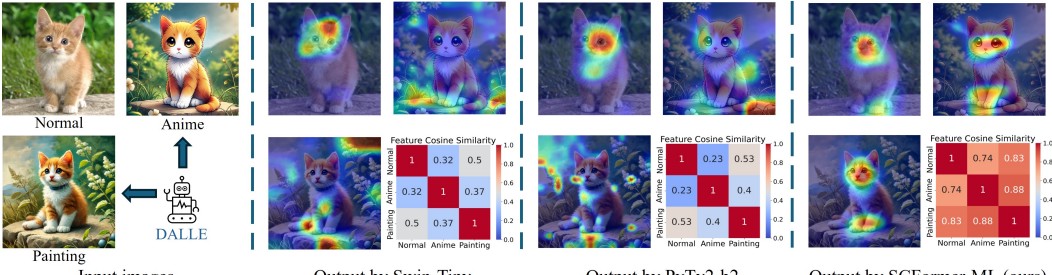

Figure 6: Grad-CAM activation maps on one trained image (normal) and two unseen OOD images (anime, painting) generated by DALLE, alone with the feature cosine similarity matrix.

Fig. 6 illustrates that all networks can identify the pattern associated with the cat in the trained image ("normal cat"). However, both Swin and PvTv2 were unable to locate the correct patterns in the two unseen OOD images, which is also evident in their low feature similarity scores across the three images. In contrast, the proposed SCFormer demonstrates a strong insensitivity to style variations, consistently identifying critical features (such as the face and feet) across all three images and exhibiting high feature similarity even in the presence of stylistic differences. These key metrics underscore the superior capacity of SCFormer in learning robust and stable visual representations.

## 5 CONCLUSION

This paper focuses on improving the robustness-efficiency trade-off of lightweight vision architectures. By targeting the channel width and the abundance of spatial conditions behind it are vital for robustness, we highlight that current token mixers, by overly focusing on token-wise exchanges, limit spatial condition representations and thus require assigning more channels to maintain robustness. To this end, we propose the spatial coordination attention (SCA) that enriches the feature representation boundary via learning attention correspondence across spatial maps with diverse pixel connectivity. By enlarging the representation boundary during token mixing, the proposed SCA can achieve robust visual modeling with fewer channels, thus improving the efficiency-robustness trade-off. Integrating SCA with our hybrid designs, SCFormers emerges as a cutting-edge prototype, exhibiting superior robustness, efficiency, and accuracy across a wide range of vision tasks.

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

# A APPENDIX

In the appendix, we offer additional information and discussions regarding:

## A.1 DETAILED ILLUSTRATION OF KERNEL VISUALIZATIONS

Figure 2 in the main text showcases the kernel property of two existing efficient SAs alongside the proposed SCA. This visualization employs the Spatial Fourier Spectrum map, the channel-wise mutual information correlation map, and the channel-wise kernel distribution map. In the following, we elaborate on their individual principles and our implementations.

**The Fourier spectrum** visualizes the frequency components captured by the SA operator, which is vital in understanding how variations in pixel intensities occur across different scales. High-frequency components capture fine details and edges (the center of the spectrum), revealing textures and sharp transitions, while low-frequency components (the part away from the center) represent overall shapes and smooth variations, such as directional gradients. A balanced representation of both high and low frequencies is essential for a robust understanding of image content. Given an image tensor $\mathbf{z} \in \mathbb{R}^{c \times h \times w}$ output by a specific operator (e.g., SA), the Spatial Fourier Spectrum visualization can be made as follows:

1. Compute the Fourier transform for each channel $\{\mathbf{z}_i\}_{i=1}^c \in \mathbb{R}^{h \times w}$:

$$\mathbf{F}_i(u,v) = \mathcal{F}\{\mathbf{z}_i(x,y)\} = \sum_{x=0}^{h-1} \sum_{y=0}^{w-1} \mathbf{z}_i(x,y) e^{-2\pi \mathbf{i}\left(\frac{ux}{h} + \frac{vy}{w}\right)}, \tag{5}$$

in case we using Euler's formula:

$$e^{-2\pi \mathbf{i}\theta} = \cos(2\pi\theta) - \mathbf{i}\sin(2\pi\theta), \tag{6}$$

we can express the Fourier transform as:

$$\mathbf{F}_i(u,v) = \sum_{x=0}^{h-1} \sum_{y=0}^{w-1} \mathbf{z}_i(x,y) \left( \cos\left(2\pi\left(\frac{ux}{h} + \frac{vy}{w}\right)\right) - \mathbf{i}\sin\left(2\pi\left(\frac{ux}{h} + \frac{vy}{w}\right)\right) \right). \tag{7}$$

2. Compute the magnitude spectrum:

$$|\mathbf{F}_i(u,v)| = \sqrt{\text{Re}(\mathbf{F}_i(u,v))^2 + \text{Im}(\mathbf{F}_i(u,v))^2}, \tag{8}$$

where the real part $\text{Re}(\cdot)$ and imaginary part $\text{Im}(\cdot)$ indicates the modelling of cosine components and sine components of the input signal, respectively. They can be illustrated as:

$$\text{Re}(\mathbf{F}_i(u,v)) = \sum_{x=0}^{h-1} \sum_{y=0}^{w-1} \mathbf{z}_i(x,y) \cos\left(2\pi\left(\frac{ux}{h} + \frac{vy}{w}\right)\right),$$

$$\text{Im}(\mathbf{F}_i(u,v)) = -\sum_{x=0}^{h-1} \sum_{y=0}^{w-1} \mathbf{z}_i(x,y) \sin\left(2\pi\left(\frac{ux}{h} + \frac{vy}{w}\right)\right). \tag{9}$$

3. Average the magnitude spectrum over channels:

$$\text{Average Spectrum}(u,v) = \frac{1}{c} \sum_{i=1}^c |\mathbf{F}_i(u,v)|^2 \tag{10}$$

4. Visualization with logarithmic scaling:

$$\text{Visualized Spectrum}(u, v) = \log(1 + \text{Average Spectrum}(u, v)) \tag{11}$$

**The mutual information correlation map** compares the mutual information between each pair of channels in an image tensor $\mathbf{z} \in \mathbb{R}^{c \times h \times w}$. It reveals the information redundancy among channels based on Information Bottleneck (IB) theory. For an image tensor processed by a specific operator (e.g., SA), lower inter-channel redundancy indicates greater feature diversity and a broader pattern representation boundary within a given channel width, which statistically reduces the risk of overfitting and enhances both robustness and parameter-efficiency. We use the *sklearn.metrics.mutual_info_score* package to calculate the numerical approximation of mutual information between different channels within the same feature. The calculation and visualization process can be conducted as follows.

1. Define channel-wise mutual information:

$$I(X; Y) = \sum_{x \in X} \sum_{y \in Y} p(x, y) \log \left( \frac{p(x, y)}{p(x)p(y)} \right). \tag{12}$$

2. Compute mutual information for each channel pair $(\mathbf{z}_i, \mathbf{z}_j). \in \mathbb{R}^{h \times w}$:

$$I(\mathbf{z}_i; \mathbf{z}_j) = \sum_{x \in \mathbf{z}_i} \sum_{y \in \mathbf{z}_j} p(x, y) \log \left( \frac{p(x, y)}{p(x)p(y)} \right), \ i, j \in \{c\}. \tag{13}$$

This results in a 2D mutual information matrix:

$$\mathbf{M}_{i,j} = I(\mathbf{z}_i; \mathbf{z}_j), \quad \text{for } i, j = 1, \ldots, c. \tag{14}$$

3. The mutual information matrix ($\mathbf{M}$) can be visualized using a heat map.

**The channel-wise kernel distribution map** intuitively displays the information diversity as well as the representation boundary of an image tensor $\mathbf{z} \in \mathbb{R}^{c \times h \times w}$. It computes the stochastic neighbour embedding of $z$ to shows the relative position of patterns captured by each individual channel ($\mathbb{R}^{h \times w}$) in the feature embedding space. It can be made as follows.

1. 0-1 standardization:

$$\mathbf{z}_i'(x, y) = \frac{\mathbf{z}_i(x, y) - \min(\mathbf{z}_i)}{\max(\mathbf{z}_i) - \min(\mathbf{z}_i)}. \tag{15}$$

2. Flatten the spatial dimenssion:

$$\mathbf{Z}_{\text{flat}} = \text{reshape}(\mathbf{z}', (c, h \cdot w)). \tag{16}$$

3. Dimensionality projection using t-SNE (Van der Maaten & Hinton, 2008).:

$$\mathbf{Z}_{\text{tsne}} = \text{t-SNE}(\mathbf{Z}_{\text{flat}}, \text{n\_components} = 3). \tag{17}$$

4. 3D visualization

$$\text{Plot}(\mathbf{Z}_{\text{tsne}}[:, 0], \mathbf{Z}_{\text{tsne}}[:, 1], \mathbf{Z}_{\text{tsne}}[:, 2]). \tag{18}$$

Please note that we use the t-SNE algorithm to implement the dimensionality projection. Different algorithms (e.g., traditional PCA) may result in some differences in absolute locations.

### A.2   COMPREHENSIVE VISUALIZATIONS.

We present additional comparisons regarding the kernel properties of our SCFormer versus existing representation networks in Fig. 7. Using the same input image, we track variations in feature kernel properties in stages 1, 2, and 4 for a comprehensive perspective.

As illustrated in Fig. 7, all comparison networks retain low-level and high-frequency information to varying extents in stage 1, but typically diminish high-frequency information as the network progresses deeper into stages 2 and 4. In contrast, our SCFormer captures a more diverse range of frequency-level information in stage 1 and maintains this diversity through stage 4, resulting in a broader representation boundary for the same input. The kernel distribution visualizations further

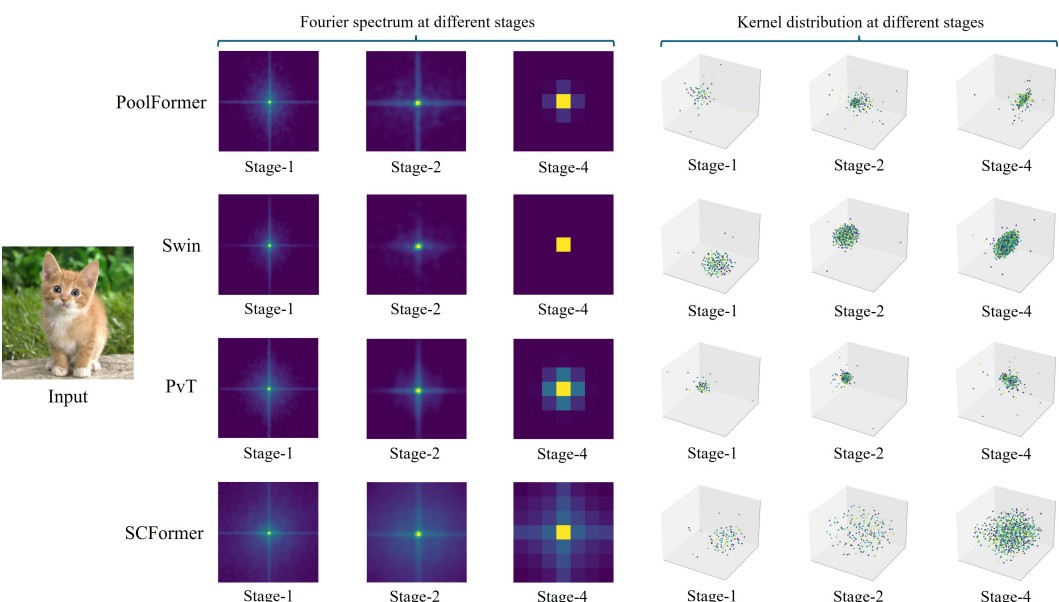

Figure 7: Overall Kernel property visualization for features output by the last token mixer at stages 1, 2, and 4. We compare our SCFormer-L with PoolFormer-S36, Swin-Tiny, and PvTv2-B2. They have similar parameter budgets.

support this observation, revealing that the compared methods often overfit to specific patterns (e.g., textures in low-frequency), leading to collapsed distributions. In contrast, SCFormer exhibits a wider kernel distribution across all stages, indicating its resilience to overfitting and its continuous effort to capture patterns with varying embedding distributions. These metrics evidently highlight the superiority of our proposed SCFormer in capturing extensive and multilevel visual cues, which are essential for attaining robust representation.

## A.3   FINE-GRAINED VISUALIZATIONS OF SCA.

In order to give deeper and more intuitive explanations about why SCA boosts the robustness, we track the feature flows during the SCA processing to visualize the property of each intermediate feature. To better remove distractions, we chose the first SCA block in the SCFormer-L as the specimen. The visualization results are presented in Fig. 8

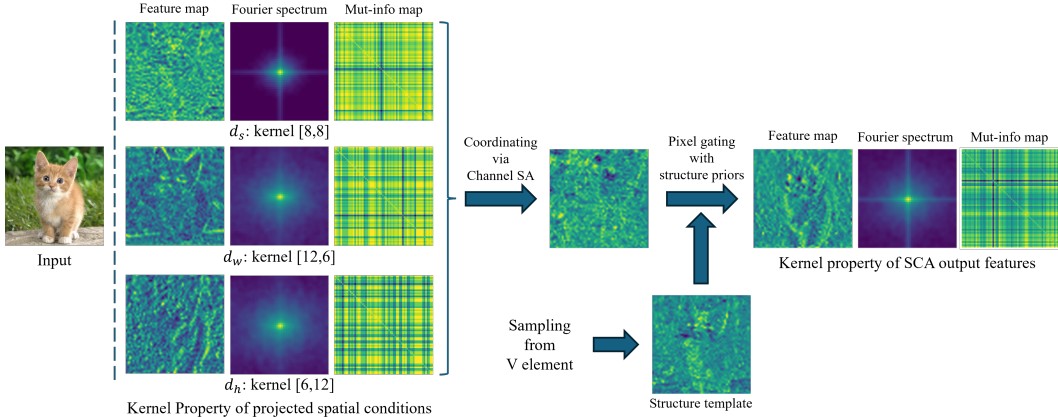

Figure 8: Fine-grained visualization dring SCA processing. We track the feature flows in the first SCA block of SCFormer-L to display the variation of feature properties.

As presented in the visualization, we can notice that the descriptor with a square border ($d_s$) tends to capture low-level information while reducing high-frequency signals. Instead, descriptors with asymmetric borders ($d_h$ and $d_w$) tend to maintain high-frequency signals. As we have verified in Fig.1 and 7 that **existing ViTs tend to has weak capacity in representing high-level frequency signals.** This weakness, in part, can be regonized by their spatial modelling are made via local descriptors with square borders before SA calculation. However, during the proposed SCA, we choose to combine multiple local descriptors with square and asymmetric borders to perform a more comprehensive spatial modeling, thus keeping the signal density and diversity for a more robust SA calculation.

Furthermore, unlike existing SA that performed in an depth-wise manner (limited in individual spatial maps), we choose to calculate the SA along channels for coordinating the spatial conditions deconstructed by different descriptors; this schema, in principle, further boosting the representation diversity to give a robust view. In the end of SCA, we employ a pixel-gating operation to fuse the coordinating results with the structure template sampled from the V element. This key process can keep the final output of SCA with semantic structures. Finally, as shown in Fig. 8, the output feature of SCA not only captured rich frequency-level information (shown in the Fourier spctrum), but also kept a low inter-channel information redundancy (shown in the mut-info map). These advantage metrics validate the capacity of SCA in extending the feature representation boundary with high information density and diversity, thus effectively boosting robustness.

## A.4 ARCHITECTURE DETAILS OF SCFORMER

The architecture configurations of six SCFormers are shown in Tab. 7. They share the same structure, but vary in depth, width, channel expansion (Exp.) ratio, and the control parameter $\tau$ for SFFN.

Table 7: Architecture configurations for six SCFormer variants.

| Module | Output Res. | Layer configuration | | SCFormer | | | | | |
|---|---|---|---|---|---|---|---|---|---|
| | | | | XXS | XS | S | M | ML | L |
| Stem | $\frac{h}{4} \times \frac{w}{4}$ | Conv 3×3, BN, GELU Conv 3×3, BN, GELU | # Channels | 24 | 32 | 40 | 48 | 64 | 72 |
| Stage-1 | $\frac{h}{4} \times \frac{w}{4}$ | SCFormer Block | # Blocks | 2 | 2 | 2 | 2 | 2 | 4 |
| | | | # Channel Exp. | 8 | 4 | 4 | 4 | 4 | 4 |
| Patch Embedding | $\frac{h}{8} \times \frac{w}{8}$ | Conv 3×3 BN,GELU | # Channels | 48 | 64 | 80 | 96 | 128 | 144 |
| Stage-2 | $\frac{h}{8} \times \frac{w}{8}$ | SCFormer Block | # Blocks | 2 | 2 | 2 | 2 | 4 | 4 |
| | | | # Channel Exp. | 4 | 4 | 4 | 4 | 4 | 4 |
| Patch Embedding | $\frac{h}{16} \times \frac{w}{16}$ | Conv 3×3 BN,GELU | # Channels | 120 | 160 | 200 | 200 | 300 | 320 |
| Stage-3 | $\frac{h}{16} \times \frac{w}{16}$ | SCFormer Block | # Blocks | 4 | 6 | 8 | 10 | 12 | 16 |
| | | | # Channel Exp. | 4 | 4 | 4 | 4 | 4 | 4 |
| Patch Embedding | $\frac{h}{32} \times \frac{w}{32}$ | Conv 3×3 BN,GELU | # Channels | 192 | 256 | 320 | 384 | 512 | 512 |
| Stage-4 | $\frac{h}{32} \times \frac{w}{32}$ | SCFormer Block | # Blocks | 2 | 2 | 2 | 2 | 4 | 4 |
| | | | # Channel Exp. | 4 | 4 | 4 | 4 | 3 | 4 |
| Head | 1 × 1 | GAP LayerNorm Linear | # Output dim. | 1000 | 1000 | 1000 | 1000 | 1000 | 1000 |
| The control parameter $\tau$ for SFFN | | | | $h*w/16$ | $h*w/8$ | $h*w/8$ | $h*w/8$ | $h*w/8$ | $h*w/4$ |
| FLOPs (G) | | | | 0.3 | 0.6 | 1.0 | 1.4 | 3.4 | 5.2 |
| Parameters (M) | | | | 2.0 | 3.9 | 6.7 | 11.8 | 22.9 | 31.4 |

## A.5 IMAGENET-1K EXPERIMENTAL SETTINGS

Detailed ImageNet-1k experimental settings for SCFormers are outlined in Tab. 8 to reproduce the performance reported in our paper. This setting is well aligned with most of the SOTA models compared (Vasu et al., 2023a; Shaker et al., 2023) in our main paper. Note that we do not use knowledge distillation in our main paper, and we offer the performance under knowledge distillation in the Appendix A.3.

Table 8: Detailed experimental settings on ImageNet-1k dataset.

| SCFormers | XXS | XS | S | M | ML | L |
|---|---|---|---|---|---|---|
| Train resolution | | | 224×224 | | | |
| Test resolution | | | 224×224 | | | |
| Train epochs | | | 300 | | | |
| Batch size | | | 2048 | | | |
| Optimizer | | | AdamW | | | |
| LR | | | 2e-3 | | | |
| LR decay | | | Cosine | | | |
| Weight decay | 0.01 | 0.015 | 0.02 | 0.025 | 0.025 | 0.05 |
| Warmup epochs | | | 5 | | | |
| Warmup schedule | | | Linear | | | |
| Label smoothing | | | 0.1 | | | |
| Dropout | | | ✗ | | | |
| Stoch. depth | ✗ | ✗ | 0.02 | 0.05 | 0.1 | 0.2 |
| Repeated Aug. | | | ✓ | | | |
| H. flip | | | ✓ | | | |
| RRC | | | ✓ | | | |
| Auto Augment | ✗ | ✗ | ✓ | ✓ | ✓ | ✓ |
| Mixup alpha | ✗ | 0.1 | 0.2 | 0.5 | 0.6 | 0.8 |
| Cutmix alpha | ✗ | 1.0 | 1.0 | 1.0 | 1.0 | 1.0 |
| Erasing prob. | | | 0.25 | | | |
| PCA lighting | | | ✗ | | | |
| Distillation | | | ✗ | | | |
| SWA | | | ✗ | | | |
| EMA deacy | | | 0.9995 | | | |
| Layer scale | | | ✗ | | | |
| Sync. BN | | | ✗ | | | |
| CE loss | | | ✓ | | | |
| BCE loss | | | ✗ | | | |
| Mixed precision | | | ✓ | | | |
| Test crop ratio | | | 0.9 | | | |

Table 9: ImageNet-1k classification accuracy under knowledge distillation.

| Model | Param (M) | FLOPs (G) | Top-1 Acc. (%) |
|---|---|---|---|
| SCFormer-XXS | 2.0 | 0.3 | 74.8 |
| SCFormer-XS | 3.9 | 0.6 | 78.7 |
| SCFormer-S | 6.7 | 1.0 | 80.9 |
| SCFormer-M | 11.8 | 1.4 | 82.0 |
| SCFormer-ML | 22.9 | 3.4 | 83.4 |
| SCFormer-L | 31.4 | 5.2 | 84.0 |

## A.6 IMAGENET-1K ACCURACY UNDER KNOWLEDGE DISTILLATION

Here, we report the performance of our SCFormer on the ImageNet-1k dataset under knowledge distillation. Specifically, we use RegNetY-16GF (Radosavovic et al., 2020) as a teacher model for hard distillation, similar to (Shaker et al., 2023; Vasu et al., 2023a). The additional settings are the same as our image classification training/testing procedure and are listed in Tab. 9. It should be noted that when using knowledge distillation, different training seeds matter to the accuracy (±0.25). We set the random seed to 0 by default. During the distillation training, we do not use the additional distillation head as (Shaker et al., 2023; Touvron et al., 2021). Instead, we use the same classification head for distillation and classification.

As shown in Tab. 9, knowledge distillation can significantly improve classification performance, which is widely used in existing attention-based backbones and some of our compared methods (Shaker et al., 2023; Maaz et al., 2022). Please note that we do not use this distillation in our main paper and report the performance here for reference.

## A.7 ROBUSTNESS EVALUATION UNDER ADVERSARIAL ATTACK

We perform robustness evaluation under adversarial attack to further validate the stability and robustness of the proposed SCormer. Specifically, we choose the classic FGSM (Goodfellow et al., 2014) and PGD (Madry et al., 2017) as two adversarial attack algorithms to attack the test samples in ImageNet-1k classification. To ensure a fair comparison with existing models, we adopt the

attack settings delineated in (Mao et al., 2022) to produce our performance. For an equitable comparison, we limit our performance comparisons to those existing models that have their adversarial attack performance officially reported in (Mao et al., 2022). We report the clean ImageNet-1k top-1 accuracy and accuracy under two types of adversarial attack algorithms in Tab. 10, respectively.

Table 10: Robustness evaluation under adversarial attack.

| Model | Params (M) | Latency (ms)↓ | ImageNet-1k Top-1 Acc. (%) | | |
|---|---|---|---|---|---|
| | | | Clean ↑ | FGSM ↑ | PGD ↑ |
| RVT-Ti (Mao et al., 2022) | 10.9 | 37.1 | 79.2 | 42.7 | 18.9 |
| SCFormer-M | 11.8 | **29.2** | **81.6** | **43.9** | **19.1** |
| Swin-T (Liu et al., 2021) | 29.0 | 90.0 | 81.3 | 33.7 | 7.3 |
| RVT-S (Mao et al., 2022) | 23.3 | 86.7 | 81.9 | 51.8 | **28.2** |
| SCFormer-ML | 22.9 | **58.4** | **82.8** | **52.2** | **28.2** |

As shown in Tab. 10, the proposed SCFormer efficiently achieves excellent robustness against adversarial attack algorithms. In particular, our SCFormer-M surpasses RVT-Ti, a model designed for defense against adversarial attacks, by achieving 1.2% and 0.2% higher top-1 accuracy under FGSM and PGD attacks, respectively. Compared with the RVT-S, the SCFormer-ML also achieves a superior robustness-speed tradeoff with significantly higher clean accuracy.

## A.8 PyTorch implementation of proposed Components

To better understand our methods and their procedures, we give the simplified Pytorch implementation codes of our proposed spatial condition coordination attention, local representation block of inception, and switchable feed-forward network in Listing 1, Listing 2, and Listing 3, respectively.

Listing 1: PyTorch implementation of the spatial Coordination attention.

```python
import torch
import torch.nn as nn

class SpatialConditionDeconstruct(nn.Module):
    def __init__(self, in_dim, ratios, pooling_proj=True, pooling_proj_rate=0.5):
        super(SpatialConditionDeconstruct, self).__init__()
        self.iter = len(ratios)
        self.poolings = nn.ModuleList()
        self.pooling_proj = pooling_proj
        self.act = nn.GELU()
        if pooling_proj:
            embed_dim = int(in_dim * pooling_proj_rate)
            self.proj = nn.ModuleList()
            self.norm_layer = nn.LayerNorm(embed_dim)

        for i in range(self.iter):
            self.poolings.append(nn.AvgPool2d(kernel_size=(ratios[i][0], ratios[i][1])))
            if pooling_proj:
                self.proj.append(nn.Conv2d(in_dim, embed_dim, kernel_size=1, stride=1, ))

    def forward(self, x):
        B, C, H, W = x.shape
        pools = []
        if self.pooling_proj:
            for i in range(self.iter):
                pool = self.poolings[i](x)
                pool = self.proj[i](pool)
                pools.append(pool.view(B, C // 2, -1))
            pools = torch.cat(pools, dim=2).permute(0, 2, 1)
            pools = self.norm_layer(pools)
        else:
            for i in range(self.iter):
                pools.append(self.poolings[i](x).view(B, C, -1))

        pools = self.act(pools)
        return pools

class SpatialCoorAtt(nn.Module):
    def __init__(self, in_dim, num_heads, pool_ratios, attn_drop=0., proj_drop=0., qkv_bias=
        True,
            pooling_proj=True,
            pooling_proj_rate=0.5):
        super(SpatialCoorAtt, self).__init__()

        self.scale = nn.Parameter(torch.ones(num_heads, 1, 1))
```

```
1026
1027        self.num_heads = num_heads
           self.head_dim = in_dim // num_heads
1028
1029        if pooling_proj:
               embed_dim = int(in_dim * pooling_proj_rate)
1030           self.qk = nn.Linear(embed_dim, 2 * in_dim, bias=qkv_bias)
           else:
1031           self.qk = nn.Linear(in_dim, in_dim, bias=qkv_bias)
1032
           self.v = nn.Sequential(nn.Conv2d(in_dim, in_dim, kernel_size=1, stride=1, padding=0,
1033               bias=False), nn.BatchNorm2d(in_dim))
1034
1035        self.attn_drop = nn.Dropout(attn_drop)

1036        self.proj = nn.Conv2d(in_dim, in_dim, kernel_size=1, stride=1)
1037
           self.proj_drop = nn.Dropout(proj_drop)
1038
           self.dconv = nn.Sequential(nn.Conv2d(in_dim, in_dim, kernel_size=3, stride=1, padding=1,
1039               groups=in_dim, bias=False), nn.BatchNorm2d(in_dim), nn.GELU(),)
1040
           self.mdp = SpatialConditionDeconstruct(in_dim, pool_ratios, pooling_proj,
1041               pooling_proj_rate)
1042
1043    def forward(self, x):
           B, C, H, W = x.shape
1044        N = H * W
1045
           qk = self.mdp(x)
1046        qk = self.qk(qk).reshape(B, -1, 2, self.num_heads, self.head_dim)
           qk = qk.permute(2, 0, 3, 1, 4)
1047        q, k = qk[0], qk[1]
1048
           v = self.v(x)
1049        v_ = v
1050
           v = v.reshape(B, C, N).permute(0, 2, 1)
1051
           v = v.reshape(B, N, self.num_heads, self.head_dim).permute(0, 2, 1, 3)
1052
1053        q = q.transpose(-2, -1).contiguous()
           k = k.transpose(-2, -1).contiguous()
1054        v = v.transpose(-2, -1).contiguous()
1055
           q = torch.nn.functional.normalize(q, dim=-1)
1056        k = torch.nn.functional.normalize(k, dim=-1)
           x = (q @ k.transpose(-2, -1).contiguous()) * self.scale
1057        x = x.softmax(dim=-1)
           x = self.attn_drop(x)
1058
           x = (x @ v).reshape(B, C, H, W)
1059        x = x + x * self.dconv(v_) # structure-prior gating
           x = self.proj(x)
1060        x = self.proj_drop(x)
           return x
1061
1062
1063
1064
```

Listing 2: PyTorch implementation of the inception local representation block.

```
1067    import torch.nn as nn
1068    import torch
1069
       class InceptionLocalRep(nn.Module):
1070        def __init__(self, in_dim, c_alpha, c_beta, c_phi):
               super(InceptionLocalRep, self).__init__()
1071           assert c_alpha + c_beta + c_phi == in_dim
               self.c_alpha, self.c_beta, self.c_phi = c_alpha, c_beta, c_phi
1072           self.conv_h = nn.Conv2d(in_channels=c_alpha,
                                       out_channels=c_alpha,
1073                                    kernel_size=(3, 1),
1074                                    stride=1,
                                       padding=(1, 0),
1075                                    groups=c_alpha,
                                       bias=False)
1076           self.conv_w = nn.Conv2d(in_channels=c_beta,
1077                                    out_channels=c_beta,
1078                                    kernel_size=(1, 3),
1079                                    stride=1,
                                       padding=(0, 1),
```

```
                               groups=c_beta,
                               bias=False)
        self.conv_s = nn.Conv2d(in_channels=c_phi,
                               out_channels=c_phi,
                               kernel_size=7,
                               stride=1,
                               padding=3,
                               groups=c_phi,
                               bias=False)

    def forward(self, x):
        x1, x2, x3 = torch.split(x, [self.c_alpha, self.c_beta, self.c_phi], dim=1)
        x1, x2, x3 = self.conv_h(x1), self.conv_w(x2), self.conv_s(x3)
        x = torch.cat((x1, x2, x3), dim=1)
        return x
```

Listing 3: PyTorch implementation of the SFFN.

```
import torch.nn as nn

class SFFN(nn.Module):
    def __init__(self, in_dim, exp_rate, drop_out_rate, mlp_bias, switch):
        super(SFFN, self).__init__()
        embed_dim = in_dim * exp_rate
        self.conv_exp = nn.Conv2d(in_dim, embed_dim, kernel_size=1, stride=1, bias=mlp_bias)
        self.conv_squ = nn.Conv2d(embed_dim, in_dim, kernel_size=1, stride=1, bias=mlp_bias)
        self.act = nn.GELU()
        self.drop = nn.Dropout(drop_out_rate) if drop_out_rate > 0 else nn.Identity()

        self.mid_conv = nn.Sequential(
            nn.Conv2d(embed_dim, embed_dim, 3, 1, 1, groups=embed_dim),
            nn.GELU(),
        ) if switch else nn.Identity()

    def forward(self, x):
        x = self.conv_exp(x)
        x = self.act(x)
        x = self.drop(x)
        x = self.mid_conv(x)
        x = self.conv_squ(x)
        x = self.drop(x)
        return x
```

## A.9 LIMITATIONS AND FUTURE WORKS

In this paper, we have explored the design of vision backbones with a focus on enhancing robustness and efficiency. Our proposed SCA and associated components demonstrate superior accuracy, efficiency, and robustness compared to existing models at lightweight scales. However, our ability to validate these designs at larger scales with more parameters is constrained by the availability of computational resources. With additional computational support, we plan to undertake thorough design and validation of larger models ($> 50M$) to further our understanding of robustness.

Currently, our architecture hyperparameters are determined empirically, which may limit the full potential of our designs to achieve optimal performance. Moving forward, we intend to dive into architecture search strategies specifically tailored for enhancing robustness. This will involve the automatic selection of architecture hyperparameters, paving the way for more sophisticated and robust vision backbone designs.

