# OpenReview forum: "SCFormer: Spatial Coordination for Efficient and Robust Vision Transformers"
_ICLR.cc/2025/Conference — ICLR 2025 Conference Withdrawn Submission_

### Official Review · Reviewer_Ldgg · 2024-10-30

**Soundness:** 4
**Presentation:** 4
**Contribution:** 3
**Rating:** 6
**Confidence:** 4

**Summary:**

To balance the trade-off between efficiency and robustness of light-weight vision backbone, this paper proposed SCFormer, a hybrid ViT architecture which introduce Spatial Coordination Attention (SCA) to capture more diverse spatial dependencies and an Inceptional Local Representation (ILR) block to enhance both locality and feature diversity. The authors conducted extensive experiments to validate the model's performance across multiple benchmarks.

**Strengths:**

1. The illustration in this paper is interesting, especially OOD samples generation workflow in the `Sec.4.5` and  fine-grained visualization of SCA in `Sec.A.1`.
2. This paper is well-written and easily-understood with comprehensive introduction and the detailed description of proposed method.
3. The proposed method is clear and demonstrates good performance on various benchmarks.

**Weaknesses:**

In `Sec.4.4`, the authors provide detailed roadmap from baseline to proposed method. However, it mainly focuses on the macro design of backbone. Micro-design ablation study (*e.g.*, SCD or pixel gating in SCA module, kernel sizes of ILR and the pooling kernel in  SCA ) are also needed.

**Questions:**

1. With regard to the most interesting design of Spatial Condition Deconstruction (SCD), I wonder if more descriptors (*e.g.*, diagonal or even dilated pattern) are utilized, could the representation be further improved? Could authors provide more discussion on this?
2. What about the training time for different size models (*e.g.* average training time for one epoch)? Could authors make a further comparison with others on this metric?

I would like to be more than willing to reconsider the rating based on authors' response to `Weaknesses` & `Questions`

---

### Official Review · Reviewer_VpM6 · 2024-10-30

**Soundness:** 3
**Presentation:** 3
**Contribution:** 3
**Rating:** 5
**Confidence:** 3

**Summary:**

This work proposes a new neural network architecture, Spatial Coordination Transformer (SCFormer), for solving the dilemma between the robustness and inference efficiency in lightweight network design. This work identifies the limited channels and spatial representation causing the robustness gap in existing lightweight transformers, and is thus motivated to propose the spatial coordination attention (SCA) to leverage various spatial maps. Extensive experiments on various datasets and tasks demonstrate the superiority of the proposed lightweight network framework.

**Strengths:**

a)	This work is well organized and easy to follow. The analysis of the weakness of existing lightweight network designs is detailed and convincing.

b)	This work proposes an Inceptional Local Representation (ILR) block to effectively enhance the feature diversity and complementary to the SCA module.

c)	This work uses various visualization tools, such as Fourier spectrum, mutual information correlation map, Grad-CAM, to help understand the internal mechanisms of the proposed network.

**Weaknesses:**

i.	This work only conducts experiments on large-scale datasets. How is the performance on the small-scale datasets, such as CIFAR-10, and Tiny ImageNet, by training from scratch or finetuning on the pretrained weights from ImageNet?

ii.	Is the proposed network effective on multi-modal applications, such as CLIP [1] and LLaVA [2]?

iii.	Which kind of pooling layer,  max pooling, average pooling, or another variant,  is used in the SCD? Was an ablation study made to make this choice?

iv.	In the SCA module, how are tokens from pooling descriptors with different areas aligned along the channel dimension, given that they produce different numbers of tokens? Could you provide more details on the implementation of this alignment process and its impact on the model's performance?

[1] Radford A, Kim J W, Hallacy C, et al. Learning transferable visual models from natural language supervision[C]//International conference on machine learning. PMLR, 2021: 8748-8763.

[2] Liu H, Li C, Wu Q, et al. Visual instruction tuning[J]. Advances in neural information processing systems, 2024, 36.

**Questions:**

See the weakness

---

### Official Review · Reviewer_5s53 · 2024-11-02

**Soundness:** 2
**Presentation:** 2
**Contribution:** 2
**Rating:** 3
**Confidence:** 4

**Summary:**

The paper introduces SCFormer, a hybrid Vision Transformer (ViT) that addresses the challenges of both efficiency and robustness, particularly in lightweight architectures. SCFormer introduces two main contributions: Spatial Coordination Attention (SCA) and the Inceptional Local Representation (ILR) block. SCA dynamically coordinates cross-spatial pixel interactions, broadening spatial representation capacity, while ILR enhances local token diversity before self-attention. The model is evaluated across multiple benchmarks, demonstrating improvements in accuracy, inference speed, and robustness against domain shifts and corrupted data.

**Strengths:**

1. The problem of achieving both robustness and efficiency in lightweight models is significant, especially for real-world applications like edge computing and autonomous systems. SCFormer’s improvements in this area are well-targeted.
2. The paper presents extensive experiments across different model types, demonstrating the effectiveness of the design.
3. SCFormer exhibits  efficiency improvements over comparable models, with lower latency and fewer parameters, while maintaining or improving robustness and accuracy.

**Weaknesses:**

Major: The manuscript lacks in-depth analysis of how ILR and the SCA improve robustness.
1. In the introduction,  key arguments need more grounding than referencing related works. For example, the statement “robustness in lightweight architectures is closely tied to channel width, with wider channels offering greater capacity to capture diverse spatial features, such as textures and frequency patterns” is overly vague and lacks detailed justification or clear evidence.
2. Section 3 focus on the design of modules without discussing how they address the robustness challenges.


Minor:
1. The introduction is unclear in its use of terms like “lightweight ViT” and “hybrid ViT” (Lines 47-53), which are used interchangeably. This section would benefit from refinement to clarify the paper's focus.
2. The Inceptional Local Representation (ILR) block appears incremental, as inception-inspired methods are well-established. While the ILR provides practical improvements, these modifications lack novelty.
3. Robustness evaluation is restricted to image classification and should be extended to more diverse tasks, such as object detection, to better substantiate SCFormer's robustness claims across applications.

**Questions:**

See weaknesses.

---

### Official Review · Reviewer_Tb3f · 2024-11-03

**Soundness:** 2
**Presentation:** 3
**Contribution:** 2
**Rating:** 5
**Confidence:** 5

**Summary:**

In this paper, the authors investigate the design of visual backbones with a focus on optimizing both efficiency and robustness. They noticed that trade-off between efficiency and robustness is particularly difficult to balance in lightweight models. They present Spatial Coordination Attention (SCA) and construct SCFormer to address these challenges. SCFormer demonstrates superior performance across multiple benchmarks.

**Strengths:**

The authors propose Spatial Coordination Attention (SCA) to broaden the representation boundary and improve robustness.
The authors conduct extensive experiments to demonstrate the effectiveness of the proposed method.

**Weaknesses:**

1.In the paper, it is unclear why Spatial Condition Deconstruction works, the experiment part can not provide support. Compared to the attention block in Restormer[1], the SCA seems additionally conduct a multi-scale enhancement when generating Query and Key. If not conduct SCD, Matmul(Q,K) also achieves interaction between all the tokens. From this perspective, the novelty of SCA is limited and its function is unclear. It suggested to illustrate why process the features at these three shapes in the SCD. More experiments and analysis are welcomed.
[1] Restormer: Efficient transformer for high-resolution image restoration.
2.The operations to enhance the feature diversity in attention-based models are well explored, such as Local Context Enhancement module in [2]. The contribution of ILR is limited. It is suggested to discuss the main difference and potential advantages of ILR compared to the existing local enhancement methods. What is the effect of other enhancement manner to the SCFormer?
[2] Biformer: Vision transformer with bi-level routing attention.
3.Some related methods [3-4] are missed for comparison. It is suggested to include TransNeXt[3] and RMT[4] in the comparison. As reported in [3], TransNeXt-T achieves better performance compared to SCFormer-L including IN1K,IN-A. A comprehensive comparison of all relevant indicators and datasets can bring a clearer understanding of the performance of SCFormer compared to these state-of-the-art methods.
[3] TransNeXt: Robust Foveal Visual Perception for Vision Transformers. CVPR 24.
[4] RMT:Retentive Networks Meet Vision Transformers. CVPR 24.
4. The method section should focus more on the core contribution. More content about the function of SCA need to be added. The content for Pixel Gating, ILR and SFFN can be reduced. It is suggested write a brief introduction of the three components and present details in the appendix.

**Questions:**

Please see weaknesses.

---

### Official Review · Reviewer_vCfV · 2024-11-07

**Soundness:** 1
**Presentation:** 2
**Contribution:** 1
**Rating:** 3
**Confidence:** 5

**Summary:**

This paper proposes SCFormer, a novel transformer based architecture to consider the trade-off between efficiency and latency.
The approach incorporates Spatial Coordination Attention and an Inceptional Local Representation block.

**Strengths:**

Emphasizing robustness and efficiency is critical for deploying modern AI systems. The work focuses on the trade off which is very important.

Results show how, for a given budget, the model improves the trade off compared to other efficient models.

The paper is very well written and easy to follow

**Weaknesses:**

The paper does a good job on the efficiency side highlighting relevant works in the literature and comparing to them. However, on the robustness side that part is lacking. There is no treatment on how robustness has been addressed in the literature nor comparisons to more relevant works.


The paper ove rfocuses on efficiency and does not compare to actual robust methods. Robustness is barely considered in the related work or experiments.
There have been a large number of efficient architectures, and some of them are discussed in the paper. There are also many papers focused on robustness, particularly understanding the benefits of transformer architectures for zero-shot noise. Those are entirely omitted in the paper. There are also recent papers explicitly addressing the trade-off between robustness and efficiency. For instance:
- Zhao et al. Fully Attentional Networks with Self-emerging Token Labeling ICCV 2023 (Robustness in transformers)
- Bair et al. Adaptive sharpness-aware pruning for robust sparse networks ICLR 2024 (Efficient-Robustness Trade-off for CNN)
- Zhou et al. Understanding The Robustness in Vision Transformers. ICML 2022.

For robustness, a quick look to Imagenet-A and imagenet R results at ICCV2023 gives STL 38.2 and 51.8 respectively while the current submission yields 27.9 and 48.7 respectively. Please, clarify

**Questions:**

The paper addresses a very relevant problem in the community for deploying neural networks. While the focus seems to be efficiency, there should also be robustness information. In particular:

1. Please explain how this paper is related to other robust approaches and how experimental results compare to them.
2. Would be great to have a more comprehensive discussion of related work on robustness in vision transformers and on jointly considering robustness and efficiency.
3. For numerical evaluation, please add comparisons to (Zhao et al. Fully Attentional Networks with Self-emerging Token Labeling ICCV 2023) and any subsequent work.
4. There is a large performance gap compared to STL (ICCV2023), so a detailed analysis on why would be helpful. The number of parameters is in the same group, so comparison should be possible)

5. Would be great to have insights on why the proposed architecture is robust to OOD (similar to what is proposed / analyzed in ICML 2022). In that way, the take home message and usefulness of the paper would increase.

---

### Note · Authors · 2024-11-22

**Comment:**

Nothing to say

**Withdrawal Confirmation:**

I have read and agree with the venue's withdrawal policy on behalf of myself and my co-authors.